# Manila Declaration on Forest and Landscape Restoration: Making It Happen

**Robin L. Chazdon** [1,2,*], **John Herbohn** [2,3,*], **Sharif A. Mukul** [2,*], **Nestor Gregorio** [2], **Liz Ota** [2], **Rhett D. Harrison** [4], **Patrick B. Durst** [5,†], **Rafael B. Chaves** [6,7], **Arturo Pasa** [8], **James G. Hallett** [9], **J. David Neidel** [10], **Cathy Watson** [11] and **Victoria Gutierrez** [12]

1   Department of Ecology and Evolutionary Biology, University of Connecticut, Storrs, CT 06269, USA
2   Tropical Forests and People Research Centre, University of the Sunshine Coast, Maroochydore DC, QLD 4558, Australia; ngregori@usc.edu.au (N.G.); lota@usc.edu.au (L.O.)
3   School of Agriculture and Food Sciences, The University of Queensland, Brisbane, QLD 4072, Australia
4   World Agroforestry (ICRAF), 13 Elm Road, Woodlands, Lusaka 999134, Zambia; r.harrison@cgiar.org
5   Food and Agriculture Organization of the United Nations (FAO), Regional Office for Asia and the Pacific, Maliwan Mansion, Phra Atit Road, Bangkok 10200, Thailand; pdurst.asiaforest@gmail.com
6   Secretariat for the Environment of the State of São Paulo, Alto de Pinheiros, São Paulo SP 05459-900, Brazil; rafaelbc.sma@gmail.com
7   Department of Ecology, Institute of Biosciences, University of São Paulo, São Paulo SP 05508-090, Brazil
8   College of Forestry and Environmental Science, Visayas State University, Baybay, Leyte 6521, Philippines; arturo.pasa@vsu.edu.ph
9   Society for Ecological Restoration, Washington, DC 20005, USA; jghallett@gmail.com
10  Environmental Leadership & Training Initiative, Yale University, New Haven, CT 06511, USA; david.neidel@yale.edu
11  World Agroforestry (ICRAF), United Nations Avenue, Gigiri PO Box 30677, Nairobi 00100, Kenya; c.watson@cgiar.org
12  Commonland, Kraanspoor 26, 1033 SE Amsterdam, The Netherlands; victoria.gutierrez@commonland.com
*   Correspondence: rchazdon@gmail.com (R.L.C.); jherbohn@usc.edu.au (J.H.); smukul@usc.edu.au (S.A.M.)
†   retired.

**Abstract:** Globally, Forest and Landscape Restoration (FLR) is gaining widespread recognition from governments and policymakers for its potential to restore key ecosystem services and to improve human wellbeing. We organized an international conference on FLR, titled—Forest and Landscape Restoration: Making it Happen, between 25–27 February 2019 in Manila, the Philippines with 139 participants from 22 countries. The Forest and Landscape Restoration Standards (FLoRES) task force also met prior to the conference, which included a field visit to a pilot community-based forest reforestation site in Biliran Island, the Philippines. Based on the three-day conference, case study presentations, and FLoRES task force meeting, we prepared the Manila Declaration on Forest and Landscape Restoration to highlight the need to support quality of FLR efforts and outcomes in the tropics. Here we provide a synthesis of the main messages of the conference, with key outcomes including the Manila Declaration on Forest and Landscape Restoration, and ways forward to make quality FLR happen on the ground.

**Keywords:** forest restoration; ecosystem services; ecological integrity; livelihood; Bonn Challenge; Sustainable Development Goals

## 1. Background

On 1 March 2019, the United Nations (UN) General Assembly declared 2021–2030 as the "UN Decade on Ecosystem Restoration". This call to action recognizes the need to massively accelerate

global restoration of degraded ecosystems to provide multiple ecosystem benefits [1]. Forest and Landscape Restoration (FLR) is a key approach to achieve this huge restoration goal, with the holistic objective of regaining ecological integrity, enhancing human well-being, and improving landscape functions in deforested or degraded landscapes. Globally, forests cover more than 30 percent of the land; they provide critical livelihood and income support for at least 1.6 billion people, and are highly degraded in many regions [2,3]. Over the past decade, FLR has received increasing attention from governments and agencies for its potential to enhance key ecosystem services and to improve human wellbeing [4,5]. The Bonn Challenge, launched in 2011 and currently involving commitments from 63 nations, pledged to restore 150 million hectares of degraded and deforested areas by 2020, which was been expanded in 2014 by the New York Declaration on Forests to 350 million hectares by 2030 [6,7]. This is the largest global initiative based on FLR approaches. Another key global initiative is the Trillion Trees Partnership [8]. An early global assessment indicated over two billion hectares of deforested and degraded land presented potential opportunities for restoration [9]. A recent study focusing on biophysical aspects, however, suggests potential for 0.9 billion hectares of tree canopy cover increase worldwide [10].

Despite the great prospect and necessity of FLR in many countries, many aspects of FLR, including governance, guidance for implementation and monitoring, and success criteria and indicators have just begun to be emphasized [11–13]. Here, it is essential to realize that FLR is a process and not a project. Timescales for implementation and deadlines for deliverables based on short-term project cycles are incommensurate with the long-term start-up requirements, slow accumulation of restoration benefits, needs for local stakeholder engagement and alliances, and focus on capacity building that are essential components of FLR. Successful implementation of FLR involves many challenges, especially those involving communities and socio-economic aspects [14–17]. Many FLR efforts fail to meet their potential or address the large-scale need despite growing funds allocated to forest restoration [18,19]. Typically, failed efforts do not adopt a landscape approach, do not effectively engage actors in local landscapes, do not build local capacity, and do not provide means for sustained financing for continuation of restoration activities [20]. FLR also suffers from a widely recognized research-practice gap in restoration. Many scientists do not communicate the results or applications of their work [21]. Likewise, practitioners do not consider how interventions and outcomes from research projects can provide essential data to advance conceptual frameworks and promote better practices more broadly [20]. Moreover, scientists and practitioners work on different teams in different locations, on paths that rarely cross.

The Forest Landscape Restoration Standard (FLoRES) task force, a voluntary informal group of scientists working on FLR, began in 2017 to discuss the need to address the quality and effectiveness of FLR implementation. One of the goals of this task force is to develop a consensus process for operationalizing the core principles of FLR, which can be adopted and tailored to guide initiation and long-term sustainability of FLR processes by stakeholders within any landscape, region, or context (see [22]; this special issue). Effective FLR practices will provide significant rewards for people and nature that share a common landscape [23]. However, initiating a FLR process often requires external financing, which presents many challenges and can be difficult to sustain [24,25].

The implementation of FLR can vary considerably in temporal scale, size, and purpose [26,27]. FLR typically involves a variety of activities that can collectively achieve multiple objectives over time ([9]; see Figure 1). Guidance is needed to ensure that the practice of FLR adheres to the core principles that set FLR apart from the business-as-usual approaches that have often failed to provide solutions to global problems of deforestation, land degradation, loss of livelihoods, food and water insecurity, and marginalization of rural peoples [22].

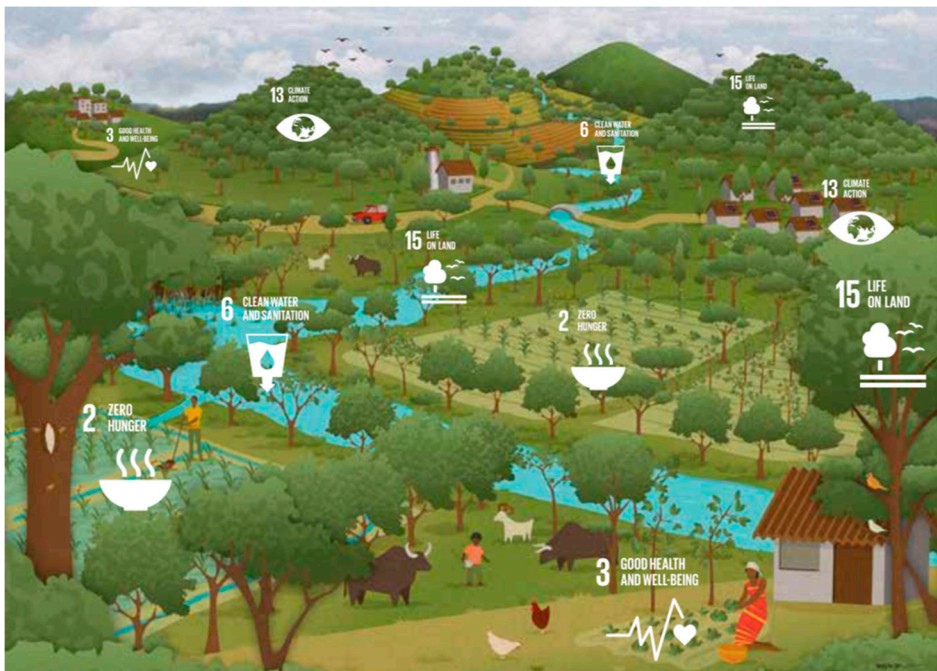

**Figure 1.** Key sustainable development goals embedded in Forest and Landscape Restoration activities (adapted from: Thaxton et al. [28] with permission from Landscapes for People, Food and Nature Initiative, 2015: Washington, DC, USA).

## 2. Key Messages and Outcomes of the FLR 2019 Conference

The FLR International Conference (www.flr2019.weebly.com) was attended by 139 participants from 22 countries from the Asia-Pacific region as well as several countries in Africa and South America (a complete list of participants can be found on the conference website). The five themes of the conference were: (i) lessons learned: Case studies of FLR design, implementation, and outcomes (positive and negative); (ii) advances in science and policy to inform FLR and improve outcomes; (iii) why operational guidelines for FLR are needed and what they might look like; (iv) engaging the private sector in FLR: From smallholders to international corporations; and (v) key interventions and directions for FLR.

The conference emphasized empirical case studies that covered successes and failures in FLR representing different tropical regions across the globe. Five of these case studies are published in this Special Issue (see [29–33]). The agenda included plenary panels, keynote addresses, and parallel sessions. Based on the keynote addresses, case studies, and panel presentations, we synthesized the following ten key messages from the conference.

(1) FLR is a dynamic and ongoing process, with no clear endpoint and no single or straight trajectory. FLR is a transformational process involving people and landscapes. Critically, it cannot be defined by projects, project budgets, and short-term outputs alone. FLR trajectories can begin in many ways, such as an ecological restoration or conservation initiative, as a local food security/agroforestry project, as a commercial forestry undertaking, research, or watershed management project. FLR can also begin through a process of collaboration and coordination within a landscape that improves local livelihoods and environmental conditions through implementing diverse types of tree cover. How it begins and how it is labelled does not matter nearly as much as the concrete steps that are made to broaden and achieve common goals and vision, regularly assess and evaluate progress, and adapt to challenges and opportunities that arise along the way.

(2)　Subnational initiatives (state, municipality, local government unit, or regional level) are the basic units of FLR activity. Centralized governments come and go. Stable and long-term support for FLR interventions should emerge from local and regional jurisdictions where turnover of staff is often lower, operations can be more focused, and agencies are invested in long-term outcomes. Furthermore, whereas local government is focused on local development, and can recognize the problems and opportunities, national government is segregated into sectorial ministries that are often competing for budgets and influence. Greater empowerment and independence should be focused at local and regional levels of government.

(3)　We need more effective ways to "learn" from experiences and refine practices to achieve greater positive impacts, to document cases of success and failure, to link specific types of interventions with specific outcomes over time, and to communicate effectively and apply this learning to new initiatives. Through case studies, we need to demonstrate how FLR can add value to rural development efforts and can help to achieve sustainable development goals.

(4)　FLR is more about people than about forests. Greater focus on local people's needs and involvement throughout the FLR process, including effective engagement in planning, operational activities, and most importantly realizing benefits and enhancing livelihoods through FLR.

(5)　Progress gained can be easily lost. Many projects fail to consider their exit strategy and simply end with no follow-up or exit plan. Building local capacity and institutions helps to build resilience and maintain forward momentum, along with continued support and guidance.

(6)　There is an urgent need to engage youth in all aspects of FLR, perhaps even linking rural activities to youth programs in nearby cities. How can younger generations see their future integrated with the landscape where they originated? What visions do they have for these landscapes and how can they be a part of this change?

(7)　The need and benefits of FLR are poorly communicated and recognized. Political and social visibility is hugely important. Many outstanding efforts deserve greater recognition and publicity.

(8)　Business models are needed that apply to FLR and that incorporate landscape approaches. These models should apply to a range of stakeholders including smallholders, local government, local businesses, and investors; and explore innovative blended financing mechanisms. The importance of cost-effectiveness needs to be elevated at project, program, and national scales.

(9)　A focus on enabling conditions is essential, particularly clear land and resource tenure, consistent supporting policies that ensure that local people derive tangible benefits from FLR initiatives. Building capacity in all its dimensions helps to establish a firm foundation for FLR initiatives. Capacity also needs to develop in government agencies to initiate cross-sectoral dialogues, coalitions, and collaborations for FLR.

(10)　Assessments of restoration success and restoration "readiness" should incorporate measures of human and social capital, community capacity, and other measures of human well-being.

To achieve effective and long-lasting outcomes, guidance and tools are needed. Working frameworks, as proposed by FLoRES (see [22]; this special issue), can help to initiate and formalize the FLR process at the landscape scale based on criteria and indicators that are tailored to the local context and based on core FLR principles [22].

One of the key achievements and outcomes of the FLR 2019 international conference was the Manila Declaration on Forest and Landscape Restoration (see Box 1), which was strongly supported by the conference participants and could be instrumental to scaling up quality FLR in the tropics. The Manila Declaration was stimulated by the leaders of the Forest and Landscape Restoration Standards task force (FLoRES; http://florestaskforce.org), who organized a workshop and field visit to a pilot reforestation project site in Biliran island, the Philippines (see [33]; Figure 2). More information about the workshop and conference are found in this blog by Cathy Watson. Discussions about conceptual and working frameworks for FLR during this workshop have been redrafted as part of this special issue [22], and provided inspiration for the Manila Declaration.

**Box 1.** Manila Declaration on Forest and Landscape Restoration.

We participants of two international meetings on the science and policy of Forest and Landscape Restoration (FLR) held in the Philippines in February of 2019:

**Recognizing** the extensive efforts that are being undertaken in support of FLR

**Aware** that implementation of FLR is expected to generate and sustain globally significant positive solutions for people, land, and climate while addressing the drivers of deforestation and forest degradation;

**Aware** that the Bonn Challenge and the New York Declaration on Forests call for the implementation of FLR across 150 million hectares by 2020 and 350 million hectares by 2030, and that current commitments by state and non-state actors have reached 170 million hectares;

**Aware** that the FLR process has reached the stage at which commitments need to be turned into practice at a scale that is commensurate with commitments;

**Aware** that the FLR process is complex, with many challenges and opportunities, requiring a holistic approach to analyse, design, implement and monitor progress as well as good governance to turn a collective vision into action at an agreed spatial and temporal scale;

**Aware** that FLR is a voluntary or policy-driven approach to change the way land and resources are being used and managed that seeks to provide socio-economic benefits, contribute to social inclusiveness and engagement of youth, women and indigenous groups, and enhance human wellbeing while increasing ecological integrity;

**Aware** that FLR is a long-term process that needs to address short-term urgent needs and requires sustained coordinated efforts;

**Observing** from numerous FLR cases that major gaps exist between the current practice of FLR and the core principles of FLR;

**Recognizing** that these gaps point to the need for more effective communication about the FLR approach and to operationalize the core principles [4,22];

**Concerned** that the practice of FLR could fail to meet expectations, e.g., by not sufficiently emphasizing the social and economic needs of local people, by poorly engaging stakeholders, by neglecting to address ecological resilience and natural regeneration potential, by compromising quality of outcomes or sustainability in an effort to rapidly reach scale, or by failing to reach a meaningful scale;

**Concerned** that both positive and negative lessons learned from interventions are essential to developing assessment metrics that integrate socio-economic and environment outcomes;

**Inspired** by the growing integration and complementarity of scientific and local knowledge regarding the effectiveness and feasibility of a wide range of restoration interventions.

**We call upon** competent and interested parties including practitioners, policymakers, civil society, public and private sectors to mobilize cross-sectoral action to implement FLR based on principles that will be made operational by actions to:

1. Develop a conceptual framework capable of guiding the practical interpretation of the FLR principles across a range of contexts along with an example of an associated working framework for a particular context;
2. Compose a short version of the working framework document, translate it into many languages, and disseminate it among many potential users;
3. Illustrate how different interventions have operationalized FLR principles (different cases can apply to each principle) through case studies and systematic reviews
4. Provide different users and sectors with targeted documentation, such as briefs and videos, to illustrate how FLR aligns with (a) broader national and international development and climate agendas; (b) good community-based governance; (c) broad environment and social policy agendas; (d) funding and investment practices and programs from both public and private sectors;
5. Encourage and support the co-development and testing of working frameworks to assess their utility and effectiveness for different FLR actors;
6. Communicate the urgency and legitimacy of FLR to different user groups, stakeholders, sectors, funding agencies, and jurisdictions;
7. Engage additional members in the FLoRES task force and broaden its interaction with interested parties;
8. Advance this agenda (steps 1–7) through international institutional platforms such as World Agroforestry (ICRAF), Asia-Pacific Network for Sustainable Forest Management (APFNet), the Global Partnership on Forest and Landscape Restoration (GPFLR), and through international conferences and workshops.

Manila, Philippines, 27 February, 2019

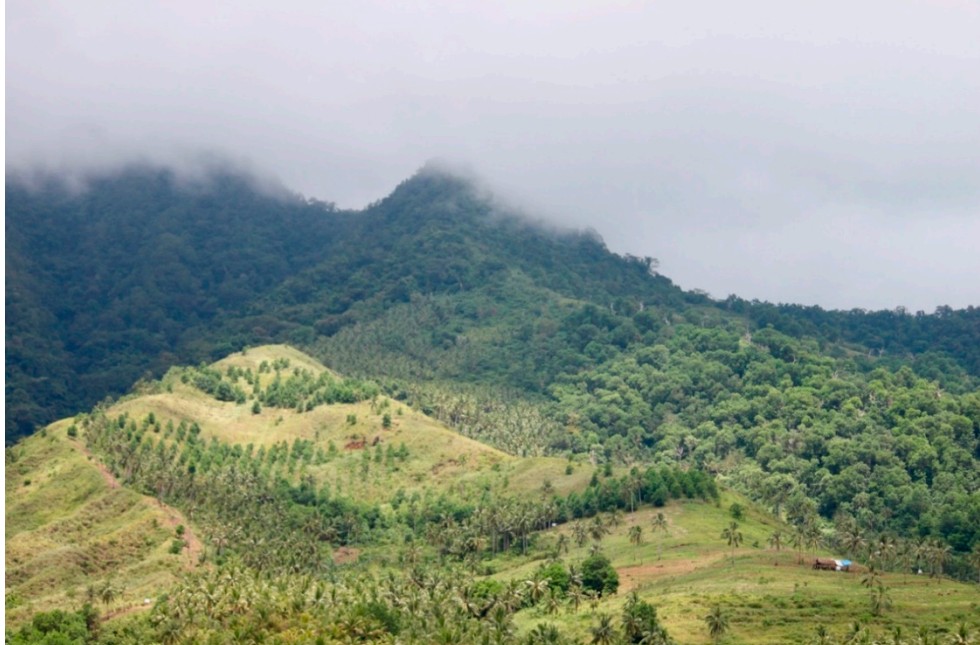

**Figure 2.** Pilot reforestation site in Biliran island, the Philippines (Photo: Sharif A. Mukul).

## 3. The Way Forward

FLR faces many challenges, and also offers many opportunities. Some of these challenges are well known and others are just starting to emerge. Some challenges are generic whereas others are more context-specific. Viewpoints differ regarding whether FLR is implemented from a project, program, or process perspective, or whether the label FLR should apply to specific types of interventions at all. Furthermore, how and why this label matters depends on how FLR is viewed by the government, non-governmental organizations (NGOs), and communities. Other conflicts arise from the increasing recognition that the short-term focus of funders (government, private sector, NGOs) is incommensurate with the long-term nature of FLR, resulting in failures and sub-optimal outcomes in many cases.

FLR is a dynamic and ongoing process and it cannot accomplish everything everywhere. Defining clear deliverables is important to engage the stakeholders and bring relevant outcomes to nature and people, but goals and targets that are unachievable can derail progress and create unrealized expectations. Practitioners and implementers need to carefully refine what is achievable under different contexts and to recognize results, even if they seem small. Some FLR interventions are top-down, government-led, whereas others are generated at local scales where the motivation comes from local people who desire and demand change. Both types of influence are important, and a major challenge is to link efforts at all of these scales to enhance the benefits and to reach the scale needed to reverse the large-scale degradation and deforestation that we are faced with. Regardless of who is initiating FLR or other forms of reforestation and restoration, the long-term success of these efforts depends heavily on the continuous engagement of local stakeholders, the fair distribution of benefits and livelihood opportunities, and adaptive management in the face of climate change and changing expectations and needs. For these reasons, co-development of frameworks to operationalize FLR and other directives of the Manila Declaration are important steps to guide future activities [22]. The key messages of the conference provide an important foundation for moving forward. In particular, we emphasize the central focus on local people's needs and involvement throughout the FLR process, including effective engagement in planning, operational activities, and most importantly realizing benefits and enhancing livelihoods through FLR.

**Author Contributions:** Conceptualization, R.L.C., V.G., J.H., S.A.M., L.O.; writing—original draft preparation, S.A.M., R.L.C., V.G., J.G.H., J.H.; writing—review and editing, S.A.M., R.L.C., V.G., J.G.H., J.H., L.O., A.P., R.D.H., P.B.D., R.B.C., J.D.N., C.W.; project administration, J.H., L.O., S.A.M., R.L.C., N.G., A.P.; funding acquisition, J.H., R.L.C., N.G., A.P. All authors have read and agreed to the published version of the manuscript.

**Funding:** The authors would like to acknowledge several funding bodies who supported the conference, including the Australian Centre for International Agricultural Research (ACIAR), Asia Pacific Forestry Network (APFNet), Forest Foundation Philippines (FFP), Philippine Council for Agriculture, Aquatic and Natural Resources Research and Development (PCAARRD), Energy Development Corporation (EDC), PARTNERS—People and Reforestation Network.

**Acknowledgments:** We extend thanks to the various conference organizing committees for all the hard work that went into making the conference a great success. The project staff and researchers from ASEM/2016/103 based at the Visayas State University worked tirelessly on the local logistics to ensure that the conference ran smoothly. We thank Lars Laestadius for helpful input on the Manila Declaration.

**Conflicts of Interest:** The authors declare no conflict of interest.

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
