# Peer review of "Manila Declaration on Forest and Landscape Restoration: Making It Happen"

_forests, doi:10.3390/f11060685_

Round 1

Reviewer 1 Report

Reviewer Recommendation and Comments for Manuscript forests-820964: Manila Declaration on Forest and Landscape Restoration: Making it happen 

The article describes the outcomes of an international conference on Forest and Landscape Restoration (FLR), and suggests some further steps to be taken for establishing FLR processes in different environments. Obviously, his is not a usual scientific article but an introduction to a special issue of the “forests” journal. Therefore, the usual criteria for an article review do not fully apply here; this has been taken into account in this review.  

General comment

The article is well and vividly written, with no objections concerning language and style. The conference summary is clearly presented, and will probably serve its introductory function well. Therefore I suggest publishing the article (under the condition that some other articles in this special edition are accepted during the review process, of course; the article would not serve as a stand-alone), possibly with some minor corrections as suggested below.

Specific comments (minor!)

  • Line 58: Please add sources for New York Declaration on Forests, and for Trillion Trees Partnership
  • L 61: It might make sense to add that Bastin et al. (2019) view the reforestation potential mainly from a biophysical perspective – a socioeconomic perspective might lead to other conclusions, i.e. a lower potential. (Only if you want, please; as this is the introduction only, I do not want to introduce unnecessary complications)
  • L 76: I guess you mean “research projects” here (rather than just “projects”)
  • L 80: As the FLoRES taskforce might not be well-known to every reader, you may want to add half a sentence for explanation in brackets (e.g. “a voluntary informal group of scientists working on FLR”, “a task force formally acting under the umbrella of….”, or the like)
  • L 113, “FLR is a journey that people and landscapes make together”: This is a vivid metaphor (which I liked very much) – however, it is also a bit strange (I never have met a landscape undertaking a journey!)
  • L 133, “it cannot be defined by…”: too apodictic; perhaps ““it cannot be defined by… alone”
  • L114: “FLR trajectories” rather than “The FLR trajectory” (since you have just argued in l. 112 that there is “no single … trajectory”); likewise in l. 117: “Such a trajectory” rather than “The trajectory”
  • Lines 112-121 in general: You argue ex negativo here, explaining what FLR is not. What is it, then?
  • Lines 135-138: Sentence incomplete (verb missing)
  • Line 146: what comes first, communicating things or recognizing them?
  • Line 152 Point missing at end of sentence
  • L 168-172: This seems very detailed here, and might possibly be skipped
  • L 178 (Figure 2): not very instructive (a map of national states somehow misrepresents the actual representation of the individual countries in the workshop; a table might do better, e.g. as it would show more clearly how many people came from each country, i.e. which regional experiences have been primarily represented).
  • Box 1: I understand this is a quotation, which cannot be changed. However, as a comment only: “the legitimacy of FLR” (point 6) is not written in stone, as this depends e.g. on the political system (and its legitimacy) in a country; one might imagine situations where FLR is not “legitimate”
  • Box 1, acronyms in point 8: Suggestion: add footnote for explanation
  • Box 1, footnote 5: Is this an original element of the declaration? If not, a literature quotation seems quite unusual for this kind of text, and might be deleted
  • L 187: this is a slight contradiction to line 112/114 where you argued that FLR is a process, not a project
  • L188: apply to what?
  • L. 184-206 in general: These are somehow very general conclusions. Would it be possible to add something more concrete, e.g. next steps to be taken by the FLoRES taskforce?  

Author Response

The article describes the outcomes of an international conference on Forest and Landscape Restoration (FLR), and suggests some further steps to be taken for establishing FLR processes in different environments. Obviously, his is not a usual scientific article but an introduction to a special issue of the “forests” journal. Therefore, the usual criteria for an article review do not fully apply here; this has been taken into account in this review.  

General comment

The article is well and vividly written, with no objections concerning language and style. The conference summary is clearly presented, and will probably serve its introductory function well. Therefore I suggest publishing the article (under the condition that some other articles in this special edition are accepted during the review process, of course; the article would not serve as a stand-alone), possibly with some minor corrections as suggested below.

Thank you for your helpful comments.

Specific comments (minor!)

  • Line 58: Please add sources for New York Declaration on Forests, and for Trillion Trees Partnership

Reference provided

  • L 61: It might make sense to add that Bastin et al. (2019) view the reforestation potential mainly from a biophysical perspective – a socioeconomic perspective might lead to other conclusions, i.e. a lower potential. (Only if you want, please; as this is the introduction only, I do not want to introduce unnecessary complications)
  • L 76: I guess you mean “research projects” here (rather than just “projects”)

Corrected.

  • L 80: As the FLoRES taskforce might not be well-known to every reader, you may want to add half a sentence for explanation in brackets (e.g. “a voluntary informal group of scientists working on FLR”, “a task force formally acting under the umbrella of….”, or the like)

Revised

  • L 113, “FLR is a journey that people and landscapes make together”: This is a vivid metaphor (which I liked very much) – however, it is also a bit strange (I never have met a landscape undertaking a journey!)

Revised. We now say “FLR is a transformational process involving people and landscapes.”

  • L 133, “it cannot be defined by…”: too apodictic; perhaps ““it cannot be defined by… alone”

Revised

L114: “FLR trajectories” rather than “The FLR trajectory” (since you have just argued in l. 112 that there is “no single … trajectory”); likewise in l. 117: “Such a trajectory” rather than “The trajectory”
Corrected.

  • Lines 112-121 in general: You argue ex negativo here, explaining what FLR is not. What is it, then?

More information about what FLR is has been added in the first paragraph of the Background section.

  • Lines 135-138: Sentence incomplete (verb missing)

Corrected.

  • Line 146: what comes first, communicating things or recognizing them?
  • Line 152 Point missing at end of sentence

Corrected.

  • L 168-172: This seems very detailed here, and might possibly be skipped

This section was shortened, but is relevant to how the Manila Declaration was developed

  • L 178 (Figure 2): not very instructive (a map of national states somehow misrepresents the actual representation of the individual countries in the workshop; a table might do better, e.g. as it would show more clearly how many people came from each country, i.e. which regional experiences have been primarily represented).

We agree the figure is not useful and it was removed

  • Box 1: I understand this is a quotation, which cannot be changed. However, as a comment only: “the legitimacy of FLR” (point 6) is not written in stone, as this depends e.g. on the political system (and its legitimacy) in a country; one might imagine situations where FLR is not “legitimate”
  • Box 1, acronyms in point 8: Suggestion: add footnote for explanation

Corrected.

  • Box 1, footnote 5: Is this an original element of the declaration? If not, a literature quotation seems quite unusual for this kind of text, and might be deleted

Yes, it was originally quoted in the declaration.

  • L 187: this is a slight contradiction to line 112/114 where you argued that FLR is a process, not a project

In many cases, FLR continues to be seen as a project, even though we do not support this view.

  • L188: apply to what?

Text now reads: Viewpoints differ regarding whether FLR is implemented from a project, program or process perspective, or whether the label FLR should apply to specific types of interventions at all.

  • 184-206 in general: These are somehow very general conclusions. Would it be possible to add something more concrete, e.g. next steps to be taken by the FLoRES taskforce?  

Please see - Chazdon et al. 2020. Co-creating conceptual and working frameworks for forest and landscape restoration based on core principles. Forests (this special issue).

Reviewer 2 Report

1. Additional references to consider:

line 65: Mansourian, S., 2016. Understanding the relationship between governance and forest landscape restoration. Conservation and Society, 14(3), pp.267-278.

line 70:

Sayer, J.A., Margules, C., Boedhihartono, A.K., Sunderland, T., Langston, J.D., Reed, J., Riggs, R., Buck, L.E., Campbell, B.M., Kusters, K. and Elliott, C., 2017. Measuring the effectiveness of landscape approaches to conservation and development. Sustainability Science, 12(3), pp.465-476.

Boedhihartono, A.K. and Sayer, J., 2012. Forest landscape restoration: restoring what and for whom?. In Forest landscape restoration (pp. 309-323). Springer, Dordrecht.   line 71:Löfqvist, S. and Ghazoul, J., 2019. Private funding is essential to leverage forest and landscape restoration at global scales. Nature ecology & evolution, 3(12), pp.1612-1615.   line 73: Hanson, C., Buckingham, K., DeWitt, S. and Laestadius, L., 2015. The restoration diagnostic. World Resources Institute (WRI). Washington DC, United States.   line 82: reference should be made to Besseau et al.   2. the introduction could be tightened. The point is to better frame the Flores work.   3. The authors state several times that FLR is not a project. It is also not about afforestation (see Veldman et al. 2015 or Bond et al. 2019) or about government-led plantation programs.. So it is unclear to me why the authors pick on projects as demonstrating what FLR is NOT, rather than focusing on what it IS.    4. The "key messages" section is very clear and informative.   5. On line 172 the authors refer to the draft White Paper but do not explain what it is.   6. The "way forward" needs more work. The sentence in line 187 is unclear. It seems to me that the key messages should form the essence of this section.  

Author Response

line 65: Mansourian, S., 2016. Understanding the relationship between governance and forest landscape restoration. Conservation and Society14(3), pp.267-278.

Reference added.

line 70: Sayer, J.A., Margules, C., Boedhihartono, A.K., Sunderland, T., Langston, J.D., Reed, J., Riggs, R., Buck, L.E., Campbell, B.M., Kusters, K. and Elliott, C., 2017. Measuring the effectiveness of landscape approaches to conservation and development. Sustainability Science12(3), pp.465-476.

Boedhihartono, A.K. and Sayer, J., 2012. Forest landscape restoration: restoring what and for whom?. In Forest landscape restoration (pp. 309-323). Springer, Dordrecht.  

References added.

line 71: Löfqvist, S. and Ghazoul, J., 2019. Private funding is essential to leverage forest and landscape restoration at global scales. Nature ecology & evolution3(12), pp.1612-1615.  

Reference added.

line 73: Hanson, C., Buckingham, K., DeWitt, S. and Laestadius, L., 2015. The restoration diagnostic. World Resources Institute (WRI). Washington DC, United States.  

line 82: reference should be made to Besseau et al.  

Reference added.

  1. the introduction could be tightened. The point is to better frame the Flores work.  

This is described in more detail in a companion paper, also published in the same special issue. The linkage is now made more clear.

3. The authors state several times that FLR is not a project. It is also not about afforestation (see Veldman et al. 2015 or Bond et al. 2019) or about government-led plantation programs.. So it is unclear to me why the authors pick on projects as demonstrating what FLR is NOT, rather than focusing on what it IS. 

Because the FLR label continues to be applied to these cases. We feel that it is as important to clarify what FLR is not as what it is.

  1. The "key messages" section is very clear and informative.  

Thank you

  1. On line 172 the authors refer to the draft White Paper but do not explain what it is.  

Revised.

  1. The "way forward" needs more work. The sentence in line 187 is unclear. It seems to me that the key messages should form the essence of this section. 

We now conclude with two additional sentences: The key messages of the conference also provide an important foundation for moving forward. In particular, we emphasize the central focus on local people’s needs and involvement throughout the FLR process, including effective engagement in planning, operational activities, and most importantly realizing benefits and enhancing livelihoods through FLR.

Reviewer 3 Report

The setup and results of the conference are generally spelled out clearly and in a way that is informative for readers.

One point that is a bit ambiguous is your stand towards FLR as project(s). In line 65/66 and in the first of your key messages (line 112 and following) you reject a conceptualization of FLR as project(s). Under the heading "The way forward" (line 187/188) however you say that there are differing viewpoints on whether FLR is best seen from o project, program, or process perspective. A more consistent approach to the question would be good.

More emphasis could be put on drivers of deforestation. Often business models that are harmful to forests are a main reason for the failure of FLR and it is very important to address these drivers of deforstation, regulate the activities of big companies and provide local people with alternative business opportunities. Drivers of deforestation are mentioned in the Manila declaration but hardly addressed in your report.

Besides these suggestions for improving the manuscript, I have noticed two passages where you should check your text:

  • In the Manila declaration, under the first "concerned" some additional words (not making any sense) have snuck into your text between "compromising quality of outcomes" and "or sustainability".
  • In line 191/192 there is a repetition of "failures"

Author Response

The setup and results of the conference are generally spelled out clearly and in a way that is informative for readers.

One point that is a bit ambiguous is your stand towards FLR as project(s). In line 65/66 and in the first of your key messages (line 112 and following) you reject a conceptualization of FLR as project(s). Under the heading "The way forward" (line 187/188) however you say that there are differing viewpoints on whether FLR is best seen from o project, program, or process perspective. A more consistent approach to the question would be good.

We recognize that in many contexts, projects are being implemented and labelled as FLR, but they do not necessarily fulfill the scope of FLR or follow the principles. We have made it clear in this paper that projects can be a starting point for FLR, but the process requires a long-term effort that could be represented by a sequence or set of coordinated projects.

More emphasis could be put on drivers of deforestation. Often business models that are harmful to forests are a main reason for the failure of FLR and it is very important to address these drivers of deforestation, regulate the activities of big companies and provide local people with alternative business opportunities. Drivers of deforestation are mentioned in the Manila declaration but hardly addressed in your report.

Besides these suggestions for improving the manuscript, I have noticed two passages where you should check your text:

  • In the Manila declaration, under the first "concerned" some additional words (not making any sense) have snuck into your text between "compromising quality of outcomes" and "or sustainability".

Corrected

  • In line 191/192 there is a repetition of "failures"

Corrected

Reviewer 4 Report

  Thank you for your summary of an important conference held in Manila, the Philippines in 2019 with the goal of advancing Forest and Landscape Restoration (FLR) globally.  Your brief article touches on many problems within FLR, as well as possibilities for achieving this goal.  I was unable to attend this conference, but now I believe that is an advantage for this review report because I can offer an outsider's perspective on points that need more clarity.

    The strengths of the article are that it succinctly summarizes a great deal of information in well-organized fashion: Background; Key Messages; and The Way Forward.  It includes a very useful reference list for further reading, as well as the full text of the "Manila Declaration," and the best info-graphic I have ever seen on SDGs embedded in FLR activities (Figure 1.). 

    The Background section offers a important single paragraph (lines 63-78) that clearly names the more serious problems of Forest and Landscape Restoration.  In particular, your sentences on the "research-practice gap" speak to me, as a practitioner of FLR in central Guatemala for the past 27 years.  You note that researchers and practitioners work "on paths that rarely cross," and they do not communicate well across sectors.  You could mention the different organizational cultures, as well:  I am repeatedly stunned by the hubris of academic researchers who ask local technicians to drive them, provide access to Mayan communities, offer translations, and rearrange work schedules-- with no offer of compensation or a token donation.   It is no wonder that smaller NGOs working in forest restoration will avoid researchers.  The good news is that grassroots organizations are cumulatively conducting more forest and landscape restoration than we have measured. The bad news is that we are not sharing experience or wisdom because the long-term focus of researchers often conflicts with the shorter-term practice focus of daily work in the field. This article returns to this important research-practice clash, and the need to engage local stakeholders in the strong conclusion (lines 189- 206).

The only content criticism this reviewer has is the confusing use of the terms "FLR principles" and "conceptual" or "working framework," especially in the Manila Declaration (Box 1).  Would it be possible to list somewhere what you mean by "FLR core principles," that are to be interpreted in the new conceptual framework?  To add to the confusion, in point 5 of the declaration, it calls for "testing of working frameworks," plural, which seems contrary to point 2 that asks for broad dissemination of the singular "working framework." 

I sincerely hope that the participants of the Manila Conference will quickly disseminate a working framework that is sensitive to the distinct cultures of researchers and practitioners and helps to bridge the gap.   

Author Response

Thank you for your summary of an important conference held in Manila, the Philippines in 2019 with the goal of advancing Forest and Landscape Restoration (FLR) globally.  Your brief article touches on many problems within FLR, as well as possibilities for achieving this goal.  I was unable to attend this conference, but now I believe that is an advantage for this review report because I can offer an outsider's perspective on points that need more clarity.

The strengths of the article are that it succinctly summarizes a great deal of information in well-organized fashion: Background; Key Messages; and The Way Forward.  It includes a very useful reference list for further reading, as well as the full text of the "Manila Declaration," and the best info-graphic I have ever seen on SDGs embedded in FLR activities (Figure 1.). 

The Background section offers a important single paragraph (lines 63-78) that clearly names the more serious problems of Forest and Landscape Restoration.  In particular, your sentences on the "research-practice gap" speak to me, as a practitioner of FLR in central Guatemala for the past 27 years.  You note that researchers and practitioners work "on paths that rarely cross," and they do not communicate well across sectors.  You could mention the different organizational cultures, as well:  I am repeatedly stunned by the hubris of academic researchers who ask local technicians to drive them, provide access to Mayan communities, offer translations, and rearrange work schedules-- with no offer of compensation or a token donation.   It is no wonder that smaller NGOs working in forest restoration will avoid researchers.  The good news is that grassroots organizations are cumulatively conducting more forest and landscape restoration than we have measured. The bad news is that we are not sharing experience or wisdom because the long-term focus of researchers often conflicts with the shorter-term practice focus of daily work in the field. This article returns to this important research-practice clash, and the need to engage local stakeholders in the strong conclusion (lines 189- 206).

The only content criticism this reviewer has is the confusing use of the terms "FLR principles" and "conceptual" or "working framework," especially in the Manila Declaration (Box 1).  Would it be possible to list somewhere what you mean by "FLR core principles," that are to be interpreted in the new conceptual framework?  To add to the confusion, in point 5 of the declaration, it calls for "testing of working frameworks," plural, which seems contrary to point 2 that asks for broad dissemination of the singular "working framework." 

I sincerely hope that the participants of the Manila Conference will quickly disseminate a working framework that is sensitive to the distinct cultures of researchers and practitioners and helps to bridge the gap.   

Thank you for your comments. The FLR principles and the conceptual and working frameworks mentioned in the Manila Declaration are the focus of a companion paper in this same special issue, so we felt it would be redundant to include them here. These links are now made as both papers will be published together in this special issue.